# What Model of Nutrition Can Be Recommended to People Ending Their Professional Sports Career? An Analysis of the Mediterranean Diet and the CRON Diet in the Context of Former Athletes

**DOI:** 10.3390/nu12123604

**Published:** 2020-11-24

**Authors:** Joanna Hołowko-Ziółek, Paweł Cięszczyk, Jarosław Biliński, Grzegorz W. Basak, Ewa Stachowska

**Affiliations:** 1Department of Human Nutrition and Metabolomics, Pomeranian Medical University in Szczecin, 71-460 Szczecin, Poland; holowkojoanna@gmail.com (J.H.-Z.); ewa.stachowska@pum.edu.pl (E.S.); 2Department of Diabetology and Internal Diseases, Pomeranian Medical University in Szczecin, 72-010 Police, Poland; 3Department of Molecular Biology, Gdansk University of Physical Education and Sports, 80-307 Gdansk, Poland; cieszczyk@poczta.onet.pl; 4Department of Hematology, Transplantation and Internal Medicine, Medical University of Warsaw, 02-097 Warsaw, Poland; hepatologia@spcsk.pl

**Keywords:** sport nutrition, elderly athletes

## Abstract

Athletes who retire from their sporting career face an increase in body weight, leading to overweight or obesity. Simultaneously, a significant number of these athletes meet the criteria of metabolic syndrome. The available literature does not offer clearly defined standards of nutrition for the discussed group of people. In this situation, it seems advisable to develop different standards of dietary behavior typical of athletes finishing their sports careers. For this purpose, the study analyzed two types of diets: the Mediterranean diet and the Calorie Restriction with Optimal Nutrition (CRON) diet based on significant calorie restrictions. Both diets seem to meet the requirements of this group of people.

## 1. Introduction

Former athletes are often people in whom the habit of practicing sports has been instilled throughout years of training. This leads to a situation that such people continue to practice the same sport discipline or another sport discipline for many more years because it results from their internal need. Other athletes, after finishing their sports careers, return to recreational sports activity only after many years in order to improve their health and well-being. There are also people who quit sports completely and do not even engage in them as a part of everyday recreation [1].

Athletes who decide to retire from sports often face an increase in body weight leading to overweight or obesity. Arliani et al. [2] observed the problem of body weight among 78% of former American football players (overweight), whereas 4% of them were classified as obese. Researchers assessed that the average increase in the Body Mass Index of these people was 2.4 kg/m^2^ since they finished their professional career [2]. This phenomenon is most often caused by continuing to consume meal portions of the same size as during periods of intense training, by lowering the energy expenditure associated with ceasing to train frequently, and by reducing the basal metabolic rate (BMR) [3,4].

Excess body weight in the group of former athletes is observed more often among people who practiced speed–strength sports than those who practiced oxygen endurance sports [5,6,7]. Excessive body weight and adipose tissue are related to the nature of the trained discipline [8,9,10]. The analysis conducted by Borchers et al. [11] indicated that obesity affects 21% of former American football players. Kujala et al. [12] observed excess body weight among former athletes who trained in strength disciplines in comparison to endurance athletes (long-distance runners, cyclists).

Components of metabolic syndrome are often observed in athletes that retire from professional sports careers. These include, among others, abnormal glycemia levels, loss of muscle mass that is replaced by adipose tissue, and the presence of nonalcoholic fatty liver disease (NAFLD) [13].

Studies conducted in the general population showed that the occurrence of cardiovascular disorders, being a part of the metabolic syndrome, increases the risk of death by 24% [14]. It is also indicated that women suffering from metabolic syndrome have a significantly higher risk of muscle weakness, also associated with cardiovascular risk factors [15]. This indicates that all activities in each group at risk of this problem are extremely important and should be implemented immediately.

The risk of metabolic syndrome in the group of former athletes was the subject of a study by Emami et al. [16]. In their work, the researchers assessed the risk of obesity, insulin resistance, and other features within metabolic syndrome in the groups of former athletes, other physically active people, and those who do not practice any sports. The study group was divided into three subgroups: active athletes (*n* = 34), retired athletes (*n* = 30), and non-sports people (*n* = 30). The obtained results indicated that abandonment of regular athletic exercise and weight cycling by power sports athletes leads to adverse outcomes as higher mean values of weight, body mass index, diastolic blood pressure, LDL-C, insulin, homeostatic model assessment (HOMA) insulin resistance (IR), and HOMA β-cell function (HOMA-%β-cell) [16].

In the athletes group, myocardial hypertrophy was observed using electrocardiography and ultrasound imaging of the heart; myocardial hypertrophy is characteristic for this period, but the heart often returns to its original state when the sports career ends [17]. The available literature indicates that regular physical activity reduces the risk of cardiovascular incidences in former athletes. Regular exercise reduces the death ratio caused by cardiovascular problems by about 31% among total physically active elderly people [18].

Sirotin and colleagues [19] showed, in a group of 122 men aged 50-59, that former athletes who practiced dynamic sports in the past are characterized by a younger biological age compared to non-training people and those athletes who practiced sports of other intensities [19].

Simultaneously, it should be mentioned that excessive weight gain is less frequent among people who maintain proper nutrition and are physically active despite ending their sports careers [6]. Endurance athletes show a lower risk of excessive body weight [6,8]. This situation may be related to the previously increased energy expenditure due to training loads. Additionally, such people are characterized by a much higher total caloric demand than people who are less active or who train in strength disciplines [20]. Maintaining proper body weight and adipose tissue content is also observed among former marathon runners; however, malnutrition is prevalent in this group during sports activity [21].

Many former athletes remain physically active after their careers [6,22]. Studies in Finland and Estonia show that more than half of former athletes regularly engage in sports in their leisure time and participate in amateur competitions. Simultaneously, such persons more often maintain normal body weight in the later years of life [23,24].

Publicly available specialist literature does not indicate clearly defined standards of nutrition dedicated to people ending their professional sports career. The Mediterranean diet and a diet based on severe caloric restrictions (Calorie Restriction with Optimal Nutrition (CRON) diet) seem to be important diets in the context of former athletes [25].

## 2. Mediterranean Diet

The term “Mediterranean diet” was first used by Ancel Keys; it refers to the diet of people who live in the Mediterranean area [26,27,28]. Numerous studies assessing the impact of this diet on improving the functioning of the body [29,30,31,32] resulted in the Mediterranean diet receiving recommendations, among others, from the European Society for the Study of Diabetes and the European Society of Cardiology (2019). These societies recommend the use of the Mediterranean diet in patients with diabetes and prediabetes with concomitant cardiovascular diseases [33]. Moreover, this diet is recommended by the Polish Society of Dietetics (2016) as a standard of dietary management in cardiology [34].

The Mediterranean diet is characterized by high consumption of vegetables (especially green leafy vegetables), cold-pressed olive oil, fruit, nuts, grains and legumes. It is recommended to eat moderate amounts of meat and fish, dairy products, and red wine. Foods such as red meat, eggs, sweets and processed foods should be eaten sporadically [35]. There are various modifications to this diet in terms of the frequency of consumption or the recommended portion size of the products. Table 1 shows a comparison of the recommended frequencies of consumption of products typical of the Mediterranean diet [36,37,38].

Dietary recommendations dedicated to the Mediterranean diet are also published in the form of a food pyramid. Such recommendations were presented in the work by Trichopoulou et al. [39,40], as shown in Figure 1.

It is recommended that the basis of the Mediterranean diet [36,37,38,39,40] be regular, moderate physical activity performed for a minimum of 30 min each day. Activities also include walking, cleaning at home, and climbing stairs. Rest is also an important part of a healthy and balanced lifestyle [37].

The next tier of the pyramid consists of liquids, which should be consumed in the amount of 1.5–2.0 L/day. Water should be the main recommended beverage, next to unsweetened herbal infusions and vegetable broths low in fat and salt [37].

The daily menu should include grain products, vegetables, fruits and reduced fat dairy products. It is recommended to use whole grain products because they contain a higher amount of dietary fiber, magnesium, and phosphorus [37]. From among vegetables, one should choose tomatoes, red pepper, eggplants, and green vegetables. They should be prepared and eaten raw, or subject to slight heat treatment with added olive oil [41]. Olive oil plays an important role in the Mediterranean food pyramid as the main source of unsaturated fatty acids. One should choose extra virgin olive oil, which is a rich source of polyphenols. In addition, extra virgin olive oil, due to its strong antioxidant and anti-inflammatory properties, is associated with a reduction in the incidence of hepatic steatosis in the elderly population with high cardiovascular risk [42,43]. The high content of polyphenols can also be found in herbs and spices, including basil, bay leaf, caraway seeds, fennel leaves, oregano, and rosemary. Bioactive compounds found in these plants show a significant effect, especially in the case of people suffering from diabetes and lipid metabolism disorders, and they contribute to reducing inflammation in the body [44].

Red wine is a characteristic element of the Mediterranean diet. It is an important source of flavonoids (tannins, anthocyanins, catechin) and stilbenes (resveratrol, tyrosyl, hydroxytyrosol) [45]. A clinical study conducted on a group of 67 men (55–75 years old) with a documented high risk of cardiovascular diseases, who were recommended to consume red wine (272 mL/day once a day), showed an improvement in glucose metabolism and the lipid profile [46]. Recommendations issued by the Mediterranean Diet Foundation inform about consuming red wine in moderation and with respect to social beliefs (recommendations for women: 14 g of alcohol a day if 1 glass/day is consumed regularly; recommendations for men: 28 g of alcohol a day if 2 glasses/day are consumed regularly) [37].

The Mediterranean diet is especially recommended for people with cardiovascular disorders. Due to the fact that former athletes often face this problem, they can use this diet both as part of dietary prophylaxis and as support to the pharmacological treatment [42,47,48]. The results of studies by de Luis et al. [49] on restrictively following the Mediterranean diet by people with cardiovascular diseases indicate that this type of diet leads to weight loss and an improvement in glucose metabolism (reduction in the blood insulin levels, insulin resistance) and in lipid parameters [49]. The Mediterranean diet is also used in the prevention and support to treatment of type 2 diabetes [50], nonalcoholic fatty liver disease [51], neurological diseases [52], rheumatic diseases [53,54], metabolic syndrome [55], cancer [56,57,58,59,60], and depressive disorders [61,62]. The aforementioned disease states are diagnosed in a significant share of former athletes, especially those who have given up practicing amateur sports [13].

Going on the Mediterranean diet contributes to the reduction of the visceral fat content, visible as a reduction of patients’ waist and hip circumference. The annual PREDIMED-Plus Trial study, involving the use of the Mediterranean diet and moderate physical activity, showed that patients experience a reduction in, e.g., waist circumference [63].

The Mediterranean diet contributes to an increase in docosahexaenoic acid (DHA) levels [64]. This acid has important functions, including impact on metabolism, regulation of gene expression, and involvement in the construction of cell walls in the body’s structures [65,66]. Omega-3 fatty acids are involved in the prevention of cardiovascular diseases [67]. The addition of DHA supplementation contributes to a reduction in the level of triglycerides in the blood plasma due to the inhibition of their resynthesis in intestinal walls, as well as the normalization of blood pressure associated with an increase in prostacyclin levels and reduced production of thromboxane A2 (TXA2) and prostaglandin E2 (PGE2) [68]. In terms of athletes, omega-3 fatty acids accelerate the regeneration of the body, which is especially important for former athletes who remain physically active [69]. DHA also affects the emotional state of a person and improves mood. Shea et al. conducted research on a group of 24 soccer players, which consisted of supplementation with DHA (3.5 g/day) for a period of 4 weeks. As a result of the intervention, perceptual and motor benefits were observed in the studied group of people [70]. The richest source of omega-3 fatty acids are oily sea fish and pharmaceutical preparations based on fish fat. They can also be found in rapeseed oil and linseed [71].

Increased levels of inflammatory mediators are often observed among people exhibiting features of metabolic syndrome. Derivatives of HETE/HODE fatty acids are the most abundant representatives of lipid peroxidation products. A study by Maciejewska et al. carried out in a group of former athletes showed a significant reduction in the level of 15S-HETE resulting from enzymatic oxidation of arachidonic acid [64]. This process is initiated by cyclooxygenase 1 (COX-1) and cyclooxygenase 2 (COX-2) [72], and it involves macrophages. The enzymes that take part in this process are characterized by the ability to oxidize phospholipids and cholesterol present in cell membranes. Oxidized forms of cholesterol esters cause pathological processes in these membranes that lead to the synthesis of pro-inflammatory cytokines. 15-HETE increases the adhesion of chemokines to endothelial cells. As a result of this process, a significant effect on monocyte circulation and the migration of molecules to the endothelial cells is observed. The 15-HETE derivative also influences the activation of NADPH oxidase, which is considered a key enzyme in the development of atherosclerosis [73]. The observed decrease in the level of 15-HETE as a result of the Mediterranean diet is related to the change in eating habits, because this diet is a rich source of omega-3 fatty acids [64].

Due to its anti-inflammatory properties, the Mediterranean diet contributes also to the reduction of uric acid concentration and the risk of hyperuricemia [74]. It is also recommended for people with a history of neoplastic disease, e.g., breast cancer [75]. Interestingly, the Mediterranean diet enriched in coenzyme Q10 modifies the expression of pro-inflammatory and endoplasmic stress-related genes in the reticula in older men and women, which also confirms its anti-inflammatory character [76]. In the context of the improvement of carbohydrate metabolism, the anti-inflammatory properties of the Mediterranean diet are also used. The available literature indicates a beneficial effect of lowering HbA1c among diabetics following the Mediterranean diet [77].

## 3. Calorie Restriction with Optimal Nutrition Diet (CRON Diet)

The CRON diet is a way of nutrition that uses the same transcription factors and proteins that are activated during intense muscle work [78]. The results of scientific studies indicate a positive effect of this diet in the context of weight reduction, improvement in glucose metabolism and lipid profile, as well as lowering blood pressure, i.e., precisely those parameters that are sometimes disturbed in former athletes [79,80,81].

Research conducted for nearly 80 years shows that the simultaneous choice of a diet based on calorie restriction and wholesome food products leads to a longer life. Analyses are conducted both among humans and animals [82,83]. The first study that analyzed the effects of an increased-restriction diet on health and life expectancy was the observation of eating habits of people living on the Japanese island of Okinawa. Among its inhabitants, there is an over 4–5 times higher percentage of people who are over 100 years of age. Simultaneously, increased life expectancy is accompanied by lower mortality from cancer and diseases of the cardiovascular system. According to the researchers, the observed situation is a consequence of the reduced-calorie diet used in the population. Compared to people living in other regions of Japan, Okinawa inhabitants consumed on average 17% less calories of energy. It is also noteworthy that the calorie restriction was maintained for the rest of life [84].

The Comprehensive Assessment of Long-term Effects of Reducing Intake of Energy (CALERIE) was a specially designed study that investigated the use of calorie restriction in humans. A combination of calorie-restricted diet with optimal nutrition was also analyzed [85,86]. The obtained results confirmed the results from animal studies, and they showed improvement in the metabolic functions of the body of people on this diet. Among participants, the researchers observed an improvement in carbohydrate metabolism parameters, i.e., improved insulin sensitivity and glucose tolerance, an improvement in lipid metabolism characterized by a reduction in the level of LDL cholesterol, as well as a reduction in the level of C-reactive protein in the blood [87,88]. Among people on the CRON diet (caloric restrictions of 30% of the total daily energy expenditure (TDEE), a reduction in the incidence of type 2 diabetes risk factors was observed, including a reduction of the body weight and adipose tissue located mainly in the abdomen, a diet based on complex carbohydrate sources, lowering blood pressure, improved lipid metabolism, and reduced incidence of atherosclerosis and other cardiovascular diseases [89].

Other authors also point to the beneficial effects of calorie restriction on health improvement. In addition to reducing the risk factors described above, studies were published that indicate a reduction in the risk of cancer or neurodegenerative diseases [86,90,91]. Calorie restriction also induces metabolic changes that occur in cells: they reduce the size of adipocytes [92], modify the secretion of adipokines, contributing to the reduction of the concentration of inflammatory mediators, and inhibit the possibility of the development of a pro-inflammatory phenotype in white adipose tissue [93,94].

A two-year study with the use of calorie restrictions of 25%, carried out in a group of 218 people with normal body weight (without obesity) and characterized by moderate physical activity, showed a significant reduction in triglyceride levels [95]. The reduction in the level of this parameter, as well as liver fatness due to the implementation of 25% calorie restriction, was also observed in Larson-Meyer’s study. The intervention was performed among obese people and lasted six months. During the study, participants were asked to maintain moderate physical activity [96].

Restrictive diets used among physically active people should be monitored in terms of nutrient deficiencies. The available literature indicates that such diets contribute to lowering the level of water-soluble vitamins. A study by Manore et al. [97] indicates a reduced supply of thiamine, riboflavin, and pyridoxine and their negative impact on the body’s efficiency. The conducted intervention showed that reduced intake of these vitamins helps to limit the maximum minute flow of oxygen (VO_2max_) and to increase the content of lactic acid in the blood, which results in a reduced ability to perform work, especially the maximum one. This situation is also unfavorable to former athletes who remain physically active [97]. An analysis by Pons et al. [98] shows that the use of calorie restriction at the level of 33% among active people helps to reduce the level of micronutrients, especially niacin and cobalamin [98]. In people practicing sports, pyridoxine is used to produce energy during exercise [99]. Pyridoxal, one of the forms of vitamin B_6_, is involved in the metabolism of lipids and carbohydrates [100], the process of gluconeogenesis and glycogenolysis [101] and is responsible for the maintenance of the proper level of glucose in the blood [102].

As already mentioned, a diet based on significant calorie restrictions also contributes to the extension of life. The first studies in this direction were carried out during the assessment of the life cycle of the nematode *Caenorhabditis elegans*. These analyses explained the molecular mechanism of life extension caused by calorie restriction (CR) in the diet. During the calorie restrictions applied in C. *elegans*, it was observed that nematodes enter the dauer phase, in which they can multiply the length of their standard life. At the cellular level, the initiation of the dauer phase is controlled by a signaling pathway associated with insulin-like growth factor (IGF-1), insulin, and also the gene transcription factor DAF-16 [103,104]. It was shown that in the situation of calorie shortages, DAF-16 enhances the expression of genes of proteins regulating oxidative stress (antioxidant enzymes) and proteins that protect the nematodes against pathogens [103,104,105]. The human DAF-16 homologues include FOX factors (*forkhead box*): FoxO1 (*forkhead box protein O1*), FoxO3, FoxO4, and FoxO6 [106,107,108,109]. FoxO is responsible for modulating the expression of genes influencing the course of life processes, such as controlling the rate of lipid, protein, and glucose synthesis. Inactivation of FoxO is observed in the liver and muscles, contributing to the repression of the regulatory genes of gluconeogenesis: phosphoenolpyruvate carboxykinase and glucose-6-phosphatase. FoxO is also a key pro-apoptotic protein and an important regulator of cell protection against oxidative stress. The cell uses this function particularly often during sports training [110]. Moreover, factor FoxO stimulates cell apoptosis by inhibiting the synthesis of Bcl-XL protein (*B-cell lymphoma-extra large*), which is a factor that protects the cell against death [111].

An important role in the control of FoxO activation is played by sirtuins (SIRT) and nicotinamidase (PNC-1; *pyrazinamidase/nicotinamidase-1*). Sirtuins are enzymes involved in the regulation of gene expression. They function as repressors of genes influencing adipogenesis and fat storage [112].

Seven isoforms of sirtuins have been defined among mammals [112]. In the context of calorie restrictions, isoform SIRT1 appears to be the most interesting, because it is the only one that is sensitive to the calorie restriction used in humans [113]. SIRT1 is found in the cell nucleus, and it regulates the transcription of FoxO and transcription factors PPARγ (*peroxisome proliferator-activated receptor γ*), NF-κB, and p53 protein [114,115].

Increased expression of the SIRT1 gene can be observed in the case of hunger or food restriction. In the case of short-term hunger, the factor activated by cyclic cAMP response element-binding protein (CREBP) is shifted to the nucleus by glucagon. This is where it binds to the promoter of the gene SIRT1 and enhances its expression. It should also be mentioned that the high activity of protein kinase A (PKA) blocks translocation of the carbohydrate-responsive element-binding protein (ChREBP) from the cytoplasm to the cell nucleus and inhibits the inhibitory effect on the expression of gene SIRT1. The inhibition of the effect can be observed in the case of fasting that lasts over 24 h. The probable cause is the present deacetylation and the concomitant blockage of CREBP by SIRT1. When food is provided, one can observe an increase in the level of glucose in the cells, rapid activation of ChREBP, and its transport to the nucleus of the cell, where the expression of gene SIRT1 is suppressed [113]. It was also proved that activation of sirtuins primarily depends on a reduction in insulin concentration induced by calorie restriction. Physical exercise can also be a sirtuin activator, which is important in the context of athletes [116,117,118].

The available literature also identifies other chemical compounds that work similarly to the sirtuin activators. They include resveratrol (3,5,4′-trihydroxystiliben), which increases the activity of human sirtuin (SIRT1) about fifteenfold. Resveratrol is a component of red wine and grape juices produced using traditional methods [119].

## 4. Application of the Mediterranean and the CRON Diet in a Group of Former Athletes

The available literature does not provide much research on the analyzed topic. As our studies have shown [13,120], reducing the Mediterranean diet used by former athletes has a positive effect on anthropometric parameters, which include reduction of the skin thickness and fat folds above the iliac crest, improvement of the nutritional status (increased arm circumference and muscle mass), and a reduction in waist circumference and body fat [120]. Another study conducted by our research team [64] with former athletes showed that the use of a diet based on approximate 500 kcal restrictions led to an increase in the content of fatty acids exhibiting anti-inflammatory character. Additionally, this diet type contributed to a significant reduction of pro-inflammatory fatty acid derivatives [64]. Another study by Hołowko et al. showed that the use of the Mediterranean diet and the CRON diet in former athletes led to an improvement in lipid and carbohydrate metabolism [25]. In this study, both calorie restriction diets, CR I (reducing daily caloric intake by 20% of total daily energy expenditure (TDEE)) and CR II (reducing daily caloric intake by 30% of TDEE), when adjusted to the caloric needs of a participant, helped to improve BMI. In former athletes who applied the reducing daily caloric intake by 30% of TDEE and lowered their body mass by 1.5–2.5 kg, 2.5–3.0 kg, and over 3.0 kg, a significant improvement in lipids (total cholesterol, LDL, and triglycerides), insulin level, and homeostatic model assessment insulin resistance (HOMA-IR) was observed. Both diets were also effective in improving the levels of leptin and adiponectin in obese former athletes [25]. Similar results were also obtained by Gawęcka et al. [121]. Interestingly, the use of caloric restrictions of the order of -800 kcal by former athletes also leads to the activation of SIRT1 responsible for the life extension process [120]. In terms of SIRT5, the result was reversed—the Mediterranean diet had a bigger influence on the increase of the expression of this sirtuin. The activation of the FoxO1 transcription factor was observed more prominently after the introduction of the Mediterranean diet.

Based on our last research study, the Mediterranean diet has a more positive influence in terms of anthropometric changes: higher body mass reduction, a decrease of skin-fat fold thickness above the hip bone, decreased waist circumference and fat tissue content, and an improvement in the nutritional status calculated through arm muscle circumference and muscle mass content [120].

As both diets have their positive sides to improving and maintaining health, the individual approach and specific needs of former athletes may outweigh the choice of one of them. This personalization of the dietary intervention may be the best way to better dietary adjustment, depending on the former athletes’ needs. More research is needed to tailor diets to the needs of former athletes, but our experience so far shows that diets that are effective in the general population (as detailed above) are also useful in maintaining normal metabolic parameters in former athletes [13,25,64,120]. Moreover, due to long-term exercise in the past, these diets also exert additional therapeutic effects on former athletes. This approach seems reasonable and research on larger groups may give even more statistically significant results.

## 5. Conclusions

Both the Mediterranean and CRON diets seem to be appropriate for athletes ending their sports careers. The available literature shows that both analyzed diets contribute to the improvement in the results of biochemical tests in a comparable way, especially lipid and carbohydrate metabolism. Moreover, the Mediterranean diet rich in omega-3 fatty acids has an anti-inflammatory character, strengthens the processes of extinguishing inflammation, and improves the anthropometric parameters in a more positive way. The CRON diet significantly contributes to the activation of SIRT1, which is associated with the process of extending life.

## Figures and Tables

**Figure 1 nutrients-12-03604-f001:**
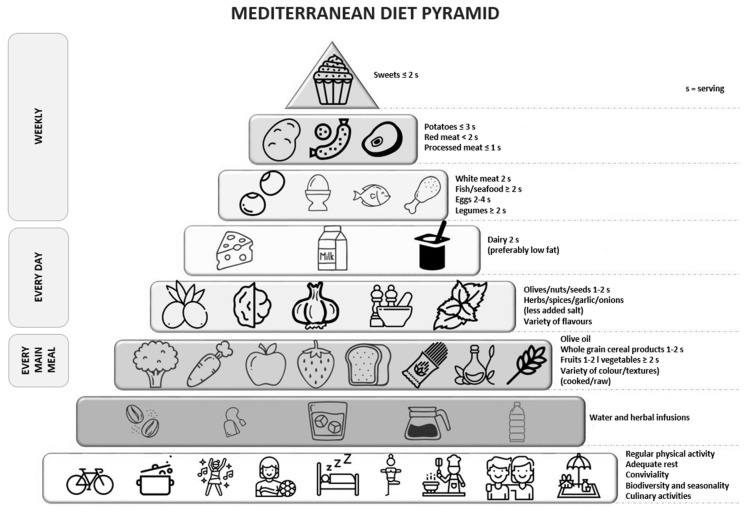
Mediterranean diet pyramid. Source: own study based on [40].

**Table 1 nutrients-12-03604-t001:** Recommended portion sizes and the frequency of consumption of products recommended in the Mediterranean diet.

Ingredients of the Diet	Make Every Day Mediterranean: An Oldways 4-Week Menu Plan Book (2019)	Mediterranean Diet Foundation (2011)	Greek Dietary Guidelines (1999)
Olive oil	in every meal	in every meal	main fat added
Vegetables	in every meal	≥2 portions in each meal	6 portions a day
Fruits	in every meal	1–2 portions in each meal	3 portions a day
Grains	in every meal	1–2 portions in each meal	8 portions a day
Legumes	in every meal	≥2 portions in each meal	3–4 portions per week
Nuts	in every meal	1–2 portions every day	3–4 portions per week
Fish/seafood	often, at least twice a week	≥2 portions per week	5–6 portions per week
Eggs	moderate portions, daily up to once a week	2–4 portions per week	3 portions per week
Poultry	moderate portions, daily to once a week	2 portions per week	4 portions per week
Dairy products	moderate portions, daily to once a week	2 portions a day	2 portions a day
Red meat	less often	<2 portions per week	4 portions per month
Sweets	less often	<2 portions per week	3 portions per week
Red wine	moderately	with moderation and respect to social beliefs	every day but in moderation

The portion size is defined as: 25 g of bread, 100 g of potatoes, 50–60 g of cooked pasta, 100 g of vegetables, 80 g of apple, 60 g of banana, 100 g of orange, 200 g of melon, 30 g of grapes, 1 glass of milk or yogurt, 1 egg, 60 g of meat, 100 g of boiled beans. Source: own study based on [36,37,38].

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
