# Peer review of "What Model of Nutrition Can Be Recommended to People Ending Their Professional Sports Career? An Analysis of the Mediterranean Diet and the CRON Diet in the Context of Former Athletes"

_nutrients, 2020, doi:10.3390/nu12123604_

Round 1

Reviewer 1 Report

The authors selected and explained two types of diets but no comparison, which makes less cohesive as one journal article. Despite the title indicates the diet for highly trained athletes who are ending or just ended their career, the contents are not specific to such athletes. From the title, I expected the clinical research, not review. The concept of this article would be a good introductory section of clinical research, such as pre-post comparison of parameters using two different diets as dependent variable.

Author Response

RESPONSES TO THE REVIEWERS’ COMMENTS

We wish to thank you all for your constructive comments in this review. Your comments provided valuable insights to refine our manuscript. In this document, we try our best to explain the reported problems.

Reviewer #1:

The authors selected and explained two types of diets but no comparison, which makes less cohesive as one journal article. Despite the title indicates the diet for highly trained athletes who are ending or just ended their career, the contents are not specific to such athletes. From the title, I expected the clinical research, not review. The concept of this article would be a good introductory section of clinical research, such as pre-post comparison of parameters using two different diets as dependent variable.

Thank you for this comment. We believe our work is suitable as a review publication. In our work, we wanted to focus on the former athlete, not active. Unfortunately, the attention of the sports world is focused primarily on the active athletes. Our team, however, wanted to highlight the problems of former athletes, especially regarding their diet. Our paper indicates what health problems these people struggle with and how improper diet is related to it. Unfortunately, nutritional standards for the group of former athletes have not been developed so far, and through this work we would like to indicate which diets may be appropriate for them.

Below are links to research papers published by our team from the Pomeranian Medical University in which the impact of diets on various parameters of former athletes was assessed:

Czerwińska M., Hołowko J., Maciejewska D., Wysokiński P., Ficek K., Wilk P., et al.: Caloric Restriction Diet (CR Diet) or Mediterranean Diet (MD) – Which is the Best Choice for Former Athletes? Cent Eur J Sport Sci Med 2016;13(1):23–35

Maciejewska D., Michalczyk M., Czerwinska-Rogowska M., Banaszczak M., Ryterska K., Jakubczyk K., et al.: Seeking Optimal Nutrition for Healthy Body Mass Reduction among Former Athletes. J Hum Kinet 2017;60:63–75.

Hołowko J., Michalczyk M.M., Zając A., Czerwińska-Rogowska M., Ryterska K., Banaszczak M., et al.: Six Weeks of Calorie Restriction Improves Body Composition and Lipid Profile in Obese and Overweight Former Athletes. Nutrients 2019;11:1461 doi:10.3390/nu11071461.

Reviewer 2 Report

English sentence structure is hard to read smoothly. There are times where the sentences are correct but the word choice makes it hard for the English reader to comprehend. It forces the reader to reread the sentence making it difficult to follow the author's point. 

Line 166 - Shae et al. conducted research on a group... The reference at the end of the sentence is [73] Larson-Meyer D.E. et al. Please check the references citation in the article with numbers in the reference list. My scan of the reference list did not find a Shae et al. 

Authors give a good review of CRON diet and its similarities with the health benefits of caloric restrictions but fail to provide evidence of its effectiveness in former athletes. If no evidence has been published on the topic, then authors should clearly state that their hypothesis is based on speculation. 

Authors talk about Red Wine as part of the Mediterranean Diet and how the CRON diet contributes to the activation of SIRT-1. However, the authors fail to mention research that links Red Wine with activation or SIRT-1. Red Wine is a source of resveratrol which is linked to SIRT-1 activation. 

Multiple times in the paper the authors refer to "footballers". They should make clear if they are meaning American Football players or Soccer players. Depending on the audience footballers may mean different athletes. Furthermore, this is important because there is a significant difference in body type of American Football players and Soccer players.

Author Response

Reviewer #2:

English sentence structure is hard to read smoothly. There are times where the sentences are correct but the word choice makes it hard for the English reader to comprehend. It forces the reader to reread the sentence making it difficult to follow the author's point. 

Thank you for this comment. We made a slight linguistic correction to remove typing errors. Thank you for any comments that have significantly improved the quality of our work.

Line 166 - Shae et al. conducted research on a group... The reference at the end of the sentence is [73] Larson-Meyer D.E. et al. Please check the references citation in the article with numbers in the reference list. My scan of the reference list did not find a Shae et al. 

We are thankful for this comment. The literature currently attached to the work contains the correct citation relating to the study by the team of Shae et al.

Authors give a good review of CRON diet and its similarities with the health benefits of caloric restrictions but fail to provide evidence of its effectiveness in former athletes. If no evidence has been published on the topic, then authors should clearly state that their hypothesis is based on speculation. 

To the best of authors knowledge there were no published research data on the effectiveness of CRON diet in former athletes without the researches published by our team and our assumptions are hypotheses.

Czerwińska M., Hołowko J., Maciejewska D., Wysokiński P., Ficek K., Wilk P., et al.: Caloric Restriction Diet (CR Diet) or Mediterranean Diet (MD) – Which is the Best Choice for Former Athletes? Cent Eur J Sport Sci Med 2016;13(1):23–35

Maciejewska D., Michalczyk M., Czerwinska-Rogowska M., Banaszczak M., Ryterska K., Jakubczyk K., et al.: Seeking Optimal Nutrition for Healthy Body Mass Reduction among Former Athletes. J Hum Kinet 2017;60:63–75.

Hołowko J., Michalczyk M.M., Zając A., Czerwińska-Rogowska M., Ryterska K., Banaszczak M., et al.: Six Weeks of Calorie Restriction Improves Body Composition and Lipid Profile in Obese and Overweight Former Athletes. Nutrients 2019;11:1461 doi:10.3390/nu11071461.

Authors talk about Red Wine as part of the Mediterranean Diet and how the CRON diet contributes to the activation of SIRT-1. However, the authors fail to mention research that links Red Wine with activation or SIRT-1. Red Wine is a source of resveratrol which is linked to SIRT-1 activation. 

We are thankful for this comment. We added such fragment relating to the effect of resveratrol on the activation of SIRT-1.

Multiple times in the paper the authors refer to "footballers". They should make clear if they are meaning American Football players or Soccer players. Depending on the audience footballers may mean different athletes. Furthermore, this is important because there is a significant difference in body type of American Football players and Soccer players.

We are thankful for this comment. We clarified this and clearly stated whether the authors means American Football players or Soccer players.

Reviewer 3 Report

  1. I would suggest the replacement of the term sportsmen and replace it with a more gender-neutral term.   
  2. I made additional suggestions in the attached pdf.  

Author Response

Reviewer #3:

I would suggest the replacement of the term sportsmen and replace it with a more gender-neutral term.   

We are thankful for this comment. We replaced the term sportsmen with more gender-neutral term of former athletes.

I made additional suggestions in the attached pdf.  

We are thankful for your review. We have replaced all identified errors with the reviewer's suggestions. In addition, we changed the drawing - we prepared our own drawing based on the source paper.

 We hope that our explanation will convince the Reviewers and allow us to publish our work.

Round 2

Reviewer 1 Report

Both diet plans are commonly employed by any people who are overweight/obese and/or pre-diabetic/diabetic. I don't see what points are specific related to sportsmen ending their professional sports career.

Author Response

RESPONSES TO THE REVIEWERS’ COMMENTS

We wish to thank you all for your constructive comments in this review. Your comments provided valuable insights to refine our manuscript. In this document, we try our best to explain the reported problems.

Reviewer #1:

Both diet plans are commonly employed by any people who are overweight/obese and/or pre-diabetic/diabetic. I don't see what points are specific related to sportsmen ending their professional sports careers.

Thank you for this comment. In the submitted article, we indicated that former athletes often exhibit features of the metabolic syndrome, including, among others, glycemic disturbances or abnormal body weight. Thus, the proposed diets are specifically related to sportsmen ending their professional sports careers. 

We hope that our explanation will convince the Reviewers and allow

Reviewer 3 Report

The revised version of the manuscript is acceptable. 

Author Response

We would like to thank you for your review.